# TMT-Based Proteomic Analysis of *Hannaella sinensis*-Induced Apple Resistance-Related Proteins

**DOI:** 10.3390/foods12142637

**Published:** 2023-07-08

**Authors:** Qiya Yang, Xi Zhang, Dhanasekaran Solairaj, Rouling Lin, Kaili Wang, Hongyin Zhang

**Affiliations:** School of Food and Biological Engineering, Jiangsu University, Zhenjiang 212013, China; yangqiya1118@163.com (Q.Y.); xi15612806202@163.com (X.Z.); solaibt@hotmail.com (D.S.); 13427550020@163.com (R.L.); 15981843773@163.com (K.W.)

**Keywords:** apple, *Hannaella sinensis*, proteomics, resistance

## Abstract

Studies on the molecular mechanism of antagonistic yeasts to control apple postharvest diseases are not comprehensive enough. Our preliminary investigations screened the biocontrol effect of *Hannaella sinensis*, an antagonistic yeast, and discovered its control efficacy on apple blue mold decay. However, the molecular mechanism of *H. sinensis*-induced resistance in apple has not been studied. In this study, proteins from apple treated with *H. sinensis* and sterile saline were analyzed using TMT proteomics technology. It was found that *H. sinensis* treatment induced the expressions of apple resistance-related proteins. Among the proteins in *H. sinensis*-induced apple, proteins related to plant defense mechanisms, such as reactive oxygen species scavenging, improvement of plant resistance and synthesis of resistant substances, improvement of plant disease resistance, the degradation of the pathogen cell wall, cell signaling, antibacterial activity, transport of defense-related substances, and protein processing, were differentially regulated. The results of this study revealed the underlying molecular mechanisms of *H. sinensis*-induced apple resistance at the protein level; the results also provided a theoretical basis for the commercial application of *H. sinensis*.

## 1. Introduction

Postharvest spoilage in apples caused by *Penicillium expansum* can reduce apple quality and storability, resulting in economic losses [1]. Traditionally, postharvest diseases of fruits and vegetables have been mainly controlled by chemical methods, and this type of treatment does have a good effect in reducing postharvest diseases in apples [2]. However, limitations, such as consumers’ demand for high-quality and safe products, environmental impact of chemical handling, and the emergence of fungicide-resistant pathogens, have motivated researchers to develop suitable alternatives to ensure safe and high-quality products [3,4]. Antagonistic yeasts have shown advantages in controlling postharvest diseases of fruits and vegetables and are safe, highly efficient, and low-cost. 

Induction of resistance in the host plant is one of the primary mechanisms of antagonistic yeasts for controlling postharvest diseases of fruits and vegetables [5]. Fruits and vegetables have evolved complex defense systems to overcome biotic and abiotic stresses, and various metabolites with different functional characters play an important role in plant defense. For example, various alkaloids, flavonoids, and phenolic compounds have antifungal activities [6]. Proteases, glucanases, and chitinases lyse the cell walls of pathogens and destroy the cells [7]. Signaling molecules, such as salicylic acid (SA), jasmonic acid (JA), and ethylene molecules, are deeply involved in the activation of plant defense responses, increasing the expressions of various defense-related genes [8,9,10]. Different antioxidant enzymes and non-enzymatic antioxidants scavenge reactive oxygen species (ROS) in cells and keep the redox balance [11]. The application of antagonistic yeast can induce the expressions of related genes in various defense pathways of fruits and vegetables, thereby helping fruits and vegetables resist pathogenic infection. For example, *Rhodotorula mucilaginosa* cultured with chitosan induces the differential expressions of proteins involved in plant hormone signal transduction, synthesis of resistance-related compounds, regulation of reactive oxygen species, and cell wall strengthening in strawberries [12].

Preliminary studies have shown that *Hannaella sinensis* can inhibit the incidence of blue mold in apples and the rot rate during natural storage. Our studies on the physiological mechanisms of *H. sinensis* in apples found that *H. sinensis* has a strong ability to colonize apple wounds and surfaces, thus occupying the nutrients and space of *P. expansum* and improving the resistance of the apple [13]. However, the molecular mechanisms behind *H. sinensis*-induced resistance in apples have not been systematically studied. Recently, the application of omics technology in understanding antagonistic yeast controlling postharvest diseases of fruit in terms of molecular mechanisms is increasing. Early transcriptomic analysis suggested that *H. sinensis* could induce apple disease resistance [14]. Due to the complexity and multidimensionality of gene expression, changes in gene expression at the transcriptional level do not necessarily correspond to changes at the protein level. Therefore, quantitative proteomics based on mass spectrometry (MS) is essential to understand how changes in protein abundance and the diversity of protein posttranslational modifications (PTMs) affect plant responses to external stimuli [3]. Recently, TMT-based proteomics has become a popular and powerful tool to study the differentially expressed proteins, due to its reproducibility, sensitivity, accuracy, and multiplexing capacity [15]. Therefore, we adopted TMT technology to investigate the proteomic changes in apples treated with *H. sinensis*, which systematically determine the alterations in global protein expression changes in apples after *H. sinensis* treatment. 

Understanding the molecular mechanisms behind disease resistance and identifying key proteins and genes of apples are crucial for the engineering of new apple cultivars and apple improvement strategies. Since climate change has accelerated recently and is expected to worsen in the future, we have reached an inflection point where comprehensive preparations to cope with the upcoming crisis can no longer be delayed [16]. In this study, we performed TMT-based proteomics analysis in apple fruits treated with *H. sinensis* and treated with sterile saline to explore the key differential proteins in apples treated with *H. sinensis* and to preliminarily elucidate the molecular mechanism of the *H. sinensis*-induced enhancement of apple resistance. At the same time, it laid a theoretical foundation for improving the disease resistance of apples using molecular means.

## 2. Materials and Methods

### 2.1. H. sinensis

*H. sinensis* was stored in our laboratory culture collection at −80 °C. *H. sinensis* (1 mL) was inoculated into 50 mL of NYDB medium. A total of 1 mL of the *H. sinensis* was transferred into a new 50 mL NYDB medium and cultured at 180 rpm at 25 °C for 20 h. The cells were counted with a hemocytometer, adjusted to the required concentration (1 × 10^8^ cells/mL), and used for the experiments.

### 2.2. Apple

Apples (*Malus domestica* cv. ‘Fuji’) from Yantai, Shandong, China were harvested at the commercial maturity stage and transferred to the laboratory. Fruits were collected in September 2022. The mean annual air temperature and total precipitation in this region were 12.95 °C and 713 mm. During the vegetative period (May–October), an average temperature of 21.46 °C, a total precipitation of 545 mm, and an average relative humidity of 72% were recorded in this region. The apple trees were trained by annual spring pruning performed according to the renewal method. The rates of N and K in orchard soil followed the recommendations for mature apple trees grown in high-density orchards in soils with a moderate organic matter status.

Apples with the same size, uniform maturity, and no mechanical damage were selected without commercial processing. Before the test, they were soaked in 0.2% sodium hypochlorite solution for 2 min, washed with water, and dried at room temperature for use. Apples with wounds inoculated with 30 μL of H. sinensis suspension (1 × 10^8^ cells/mL) were named as the treated group, and apples with the same amount of sterile normal saline were named as the control group. After drying at room temperature, apples were placed in plastic baskets, sealed with plastic wrap, and stored in a constant temperature and humidity incubator (20 °C, 95% humidity).

### 2.3. Protein Extraction from Apple Samples

A total of 5 μL of phosphatase inhibitor, 1 μL of protease inhibitor, and 10 μL of PMSF were added to each 1 mL of pre-cooled lysis buffer, mixed, and stored on ice for later use. An amount of 200 mg of pulp tissue at the apple wounds after 4 days post-inoculation was taken, placed in a petri dish, and washed with PBS twice. After centrifugation, 1 mL of the pre-cooled lysis buffer was added and manually homogenized in a glass homogenizer at 4 °C 15–30 times. The tissue homogenate was transferred to a 1.5 mL pre-cooled centrifuge tube, placed on ice for about 10 min, shaken vigorously 2–3 times during this period, and centrifuged, and the supernatant was transferred to a pre-cooled centrifuge tube. The protein concentration was determined using a BCA kit and detected by SDS-PAGE electrophoresis.

### 2.4. Proteolysis and TMT Labeling

Proteins in the samples were reduced with 2 mM dithiothreitol (DTT) for 30 min at room temperature and then alkylated with 20 mM iodoacetamide for 30 min. Proteins were isolated from lysates using chloroform–methanol extraction, pre-digested with Lys-C for 3 h, and then digested with trypsin overnight at 37 °C. The treated samples were acidified and desalted with Oasis-HLB cartridges. The digested peptides were reconstituted in 200 mM HEPES (pH 8.0) supplemented with 40% anhydrous acetonitrile and labeled with TMT10-plex reagent for 1 h, and 10% hydroxylamine was added to quench the reaction. Finally, TMT-labeled peptides were pooled and lyophilized.

### 2.5. High pH Reverse-Phase Separation

TMT-labeled peptides were dissolved in buffer A (A: 10 mM ammonium bicarbonate, 5% acetonitrile, pH 8.0) and loaded onto an Agilent 300 Extend C18 column. A linear gradient separation was set for 120 min, including 0 min-4% B; 75 min-65% B; 80 min-100% B; 98 min-100% B; 100 min-0% B; and 120 min-0% B, where B was 10 mM ammonium bicarbonate, 80% CAN, and pH 8 at a flow rate of 0.2 mL/min. The peptide mixture was divided into 12 fractions, and the eluate was collected into 1.5 mL tubes every 1.5 min from 5 min to 75.5 min, then acidified with 20% TFA. The fractionated peptides were dried and desalted before LC-MS/MS analysis.

### 2.6. Bioinformatics Analysis

The wiff file after the mass spectrometer was turned off was directly imported into the Proteome Discovery software for a database search, peptide and protein quantifications, and the next step of the information analysis. The apple proteome reference database used in this experiment was: GDDH13_1-1_prot.fasta (45116 protein sequences), accessed at https://www.rosaceae.org/species/malus/all (accessed on 6 July 2023). 

Data statistics and quality assessment were carried out by comparing the obtained protein data. Differentially expressed protein (DEP) screening and expression cluster analysis were performed through quantitative analysis of the proteins.

## 3. Results

### 3.1. Protein Differential Analysis

For differential expression analysis of samples with biological replicates, a *t*-test was performed, and a fold change ≥ 1.2 or ≤0.83 and a *p*-value < 0.05 were used as screening criteria. When FC ≥ 1.2, and *p*-value < 0.05, the protein expression was considered up-regulated; when FC ≤ 0.833, and *p*-value < 0.05, the protein was considered down-regulated. Compared with apple tissue treated with sterile saline, 647 proteins were differentially expressed in *H. sinensis*-treated apple tissue, including 348 up-regulated and 299 down-regulated DEPs (Figure 1).

### 3.2. Differential Protein GO Enrichment Analysis

The Combined Cluster Profiler software and GO database (http://www.geeontology.org) were used to perform functional annotations of DEPs, and the results are shown in Figure 2a. From biological process annotation analysis, differential proteins were annotated into 19 secondary categories, including the immune system process, metabolic process, response to stimulus, biological regulation, and detoxification, which might be associated with the apple resistance mechanism induced by *H. sinensis*. Under the molecular function category, the DEPs were annotated into 10 secondary categories, among which 353 differential proteins belonged to catalytic activity, and 13 differential proteins belonged to antioxidant activity. The DEPs might be involved in the induction of apple resistance by *H. sinensis*. Under the cell components category, the DEPs were annotated into 12 secondary categories, among which 157 proteins belonged to cell membranes, and 242 proteins belonged to organelles. These analyses showed that *H. sinensis* induced resistance by affecting apple cell components.

Tertiary enrichment analysis was performed on secondary pathways of the immune system process, metabolic process, response to stimulus, cell membrane, antioxidant activity, and catalytic activity, and the results are shown in Figure 2b. Among them, the response to biological stimuli, endoplasmic reticulum membrane, plasma membrane, oxidoreductase activity, and tertiary pathway of the redox process had the most enriched differential proteins, indicating that *H. sinensis* treatment induced the redox-related enzymes in apples, which could respond to stress.

### 3.3. Differential Protein KEGG Enrichment Analysis

Annotation analysis of the KEGG database was performed on the DEPs. The DEPs were annotated to 19 secondary pathways, as shown in Figure 3a. Among them, signal transduction in environmental information processing was enriched with 7 DEPs; amino acid metabolism, biosynthesis of other secondary metabolites, carbohydrate metabolism, and energy metabolism in metabolism were enriched with 178 DEPs; and environmental adaptations in organismal systems were enriched with 10 DEPs. These secondary pathways had a large number of differential proteins annotated in each primary pathway, indicating that *H. sinensis* improved apple resistance by differentially regulating various metabolic pathways in apples.

Six secondary pathways were subjected to tertiary enrichment analysis, as shown in Figure 3b. Among them, plant–pathogen interaction, amino sugar and nucleotide sugar metabolisms, starch and sucrose metabolisms, flavonoid biosynthesis, phenylpropanoid biosynthesis, isoquinoline alkaloid biosynthesis, and the MAPK signaling pathway were related to plant resistance. These results indicated that *H. sinensis* could improve apple resistance by inducing protein expression in resistance-related pathways.

### 3.4. Differential Protein KEGG Enrichment Analysis

After apples were treated with *H. sinensis*, the expressions of multiple enzymes in the jasmonic acid biosynthesis pathway were up-regulated, compared with the control group, as shown in Figure 4a. The synthesis of jasmonic acid comprised of the following stages: generation of α-linolenic acid from phosphatidylcholine was catalyzed by phosphatase; α-linolenic acid was catalyzed by lipoxygenase to synthesize 13(S)-HpOTrE; and 13(S)-HpOTrE was then catalyzed by allene oxide synthase and allene oxide cyclase (AOC) to generate 12-OPDA. After that, 12-OPDA was transported to the peroxisome and catalyzed by OPDA reductase (OPR) to generate OPC8, and OPC8-CoA ligase ligated OPC8 to coenzyme to generate OPC8-CoA. OPC8-CoA generated jasmonic acid after three performances of β-oxidation. Among them, allene oxide synthase 1 (AOS1), AOC, allene oxide cyclase 4 (AOC4), 12-oxodienoic acid reductase 2 (OPR2), 4-coumarate coenzyme A ligase 5 (4CLL5), and peroxisomal fatty acid β-oxidation multifunctional protein AIM1 (AIM1) were all up-regulated. The up-regulation of these enzymes promoted the final jasmonic acid synthesis, which further activated the defense system of the apples.

In apples treated with *H. sinensis*, the proteins in the protein-processing pathway in the endoplasmic reticulum were up-regulated, as shown in Figure 4b. The transporter SC61B, responsible for the transport of polypeptides into the lumen of the ER, was upregulated. After entering the ER lumen, proteins underwent folding under the actions of various molecular chaperones and enzymes. N-glycan-dependent and N-glycan-independent are the two main ways of protein folding, and among them, the expression of the luminal binding protein BiP5 involved in the N-glycan-independent pathway was up-regulated, and BiP5 was bound to the SC61B transposon and interacted with the newly synthesized polypeptide. The DnaJ proteins P58IPK and ERDJ3B, which were assisted in this process in a nucleotide-dependent manner, were also up-regulated. In addition, the expression of calnexin CNX1 involved in the N-glycan-dependent folding pathway was up-regulated. After the protein was fully folded, it was transported by vesicles from the ER to the Golgi apparatus. However, some misfolded proteins were degraded by the ERAD system, and the expressions of protein OS9 and transporter SC16B was up-regulated. Overall, *H. sinensis* treatment enhanced the protein-processing efficiency of the endoplasmic reticulum in apples.

In the plant–pathogen interaction pathway, some proteins were up-regulated in the immunity (PTI) triggered by PAMPs (pathogen-associated molecular patterns) in the plant innate immune system after apples were treated with *H. sinensis*, as shown in Figure 4c. The intracellular Ca^2+^ concentration was increased, and the sensor proteins that recognize Ca^2+^ signals, such as calcium-dependent protein kinase 1 (CDPK1) and calcium-binding protein (CML11), were up-regulated, resulting in a burst of ROS, which further induced hypersensitivity (HR). In addition, the expression of the mitogen-activated protein kinase, kinase 4 (MKK4), in the MAPK cascade was also up-regulated, ultimately regulating the expression of resistance-related genes.

As shown in Figure 4d, the expressions of some proteins in the flavonoid synthesis pathway were up-regulated in apples treated with *H. sinensis*. Among them, the expression of chalcone isomerase (CHI), which catalyzes the syntheses of pinocembrin, liquiritigenin, butin, and naringenin, was up-regulated. Chalcone isomerase (CHI) catalyzes the first two steps of flavonoid biosynthesis and produces precursors of different classes of flavonoids. The expression of UDP-glycosyltransferase 88A1 (UGT88A1), which catalyzes the production of phlorizin from phloretin, was up-regulated. The expression of BAHD acyltransferase (BAHD1), which catalyzes the transfer of caffeoyl units to the receptor substrate quinic acid to generate caffeoyl quinic acid, which accumulates as an antibacterial compound in plants, was up-regulated. 

### 3.5. Functional Classification of Related DEPs

According to the DEPs enriched from various databases, the key DEPs related to the participation of *H. sinensis*-induced improvement of apple resistance were screened. The functional statistics of specific proteins are shown in Table 1. According to the different functions of these DEPs, they were divided into seven categories: oxidative stress, improvement of plant resistance, improvement of plant disease resistance, pathogen cell wall degradation, signal transduction, antibacterial activity, substance transport, and protein processing.

## 4. Discussion

In this study, protein expressions of apples treated with *H. sinensis* and sterile saline were analyzed based on TMT proteomics technology. It was found that *H. sinensis* treatment induced the expression of resistance-related proteins in apples. Among them, the functions of DEPs were mainly reflected in scavenging reactive oxygen species, improving the synthesis of plant resistance and resistance substances, improving plant disease resistance, degrading pathogen cell walls, enhancing cell signal transduction ability, having antibacterial activity, improving the transport capacity of defense-related substances, and improving the processing efficiency of protein.

### 4.1. Scavenging Reactive Oxygen Species

When plants are subjected to abiotic and biotic stresses, plants regulate the expressions of various stress genes, which causes rapid accumulation of ROS, leading to cellular damage (lipid peroxidation, toxic product aldehydes accumulation, etc.), so it is important to control redox homeostasis in cells. After *H. sinensis* treatment in apples, proteins related to oxidative stress were expressed in large quantities; some proteins catalyzed the hydrolysis of toxic oxidation products, and some proteins participated in electron transfer during the redox process, thereby achieving redox homeostasis in cells. Among them, chloroplast envelope quinone oxidoreductase homolog (CEQORH) reduced the production of toxic oxidation products of chloroplast membrane lipids [17]. The acylamino acid-releasing enzyme (AARE2) enhanced plant stress tolerance by catalyzing the hydrolysis of oxidized proteins. L-ascorbate oxidase (AAO) could catalyze the formation of dehydroascorbic acid from ascorbic acid in vitro and regulate the redox state of ascorbic acid in vitro, thereby regulating the stress responses of plants [18]. Aldehyde dehydrogenases acted as aldehyde scavengers, catalyzing the irreversible oxidation of various endogenous and exogenous aromatic and aliphatic aldehydes to the corresponding carboxylic acids, indirectly scavenging cellular ROS and reducing lipid peroxidation [19]. FQR1 had quinone reductase activity and could catalyze the transfer of double electrons from NAD(P)H to substrates, protect cells from oxidative damage, and act as a detoxification protein for cells [20]. Malate dehydrogenase (MMDHI) reversibly catalyzed the oxidation of L-malate to oxaloacetate and simultaneously reduced NAD(P) to NAD(P)H, playing an important role in plant redox homeostasis [21]. Copper azurin (BCB) was involved in electron transfer during plant oxidation and oxidative stress and also participated in the glycero–inositol signaling pathway to control lipid peroxidation and actively regulate plant disease resistance [22].

### 4.2. Improve the Synthesize of Resistance-Related Substances

After apples were treated with *H. sinensis*, some proteins related to the synthesis of resistance-related substances and related proteins that could improve apple stress resistance were up-regulated. PPO expression in apples was up-regulated after *H. sinensis* treatment, and increasing PPO activity could enhance plant resistance to pathogens. PPO also caused the production of ROS, which activated signaling molecules that mediate plant resistance [23]. *H. sinensis* treatment in apples induced the up-regulation of multiple glutathione S-transferases (GSTs), a protein superfamily with diverse functions in plants. GST expression in kidney beans was induced upon infection with the pathogen *Uromyces appendiculatus*, and detoxifying toxic compounds were produced by *U. appendiculatus* [24]. Serine carboxypeptidase-like protein is an acyltransferase in the plant secondary metabolic pathway and is involved in the production of a series of structurally diverse natural compounds that can help plants resist infection by pathogens [25]. Subtilisin-like (SBT3.8) is an important key in plant defense responses and participates in the production of some bioactive peptides [26]. 

Salicylic acid-binding protein 2 (SABP2), a resistance signaling receptor for salicylic acid (SA), is involved in SA-induced resistance responses [9]. Plant lignification, which strengthens plant cell walls and forms a non-degradable physical barrier to most microorganisms, is a defensive response to pathogen infection [27]. Cinnamoyl-CoA reductase, which is an important key enzyme in the lignin synthesis pathway, was up-regulated in apples by *H. sinensis*. Flavonoids play important roles in plants [28]. After *H. sinensis* treatment, the expressions of proteins in the flavonoid synthesis pathway were up-regulated, including chalcone isomerase (CHI), UDP-glycosyltransferase 88A1 (UGT88A1), and BAHD acyltransferase (BAHD1). 

Phytohormones play an essential role in establishing signaling networks that regulate plant growth and stress-related responses, among which JAs can effectively adjust responses to environmental stress by inducing a series of gene expressions and promoting plant responses to external damage (mechanical damage, herbivore injury, and insect injury) and responses to pathogen infection [29]. After *H. sinensis* treatment, the expressions of some important enzymes in the JA biosynthesis pathway were up-regulated, including AOS1, AOC, allene oxide cyclase enzyme 4 (AOC4), 12-oxophytodienoate reductase 2 (OPR2), 4-coumarate coenzyme A ligase 5 (4CLL5), and peroxisomal fatty acid β-oxidation multifunctional protein AIM1 (AIM1).

### 4.3. Improve Plant Disease Resistance

The plant immune system senses pathogenic infection pathogens through two branches and then activates the immune response. PAMPs, such as the flagellin of bacteria and chitin of fungi, are sensed by pattern recognition receptors on the surface of plant cells, leading to PTI, which constitutes the first line of defense in plant defense responses [30]. After *H. sinensis* treatment in apples, the expressions of proteins in the PTI and ETI immune pathways were up-regulated. The sensor proteins calcium-dependent protein kinase 1 (CDPK1) and calcium-binding protein (CML11), which recognize Ca^2+^ signals, were up-regulated. The up-regulated expression of RPM1-interacting protein 4 (RIN4), which can be targeted and modified by various pathogen effector proteins, initiated plant immune ETI in R-protein-dependent or independent manners, acting at the intersection between PTI and ETI [31]. The up-regulated expression of E3 ubiquitin protein ligase (ATL6) affected the sensitivity of flg22 responses and affected plant defense responses [32].

### 4.4. Degrade Pathogen Cell Walls

When pathogens infect plants, the responses of various defense hormones, such as SA, JA, and ethylene, induce the production of pathogenesis-related (PR) proteins. Some of the PR proteins have a degrading effect on the cell wall of pathogens and launch a counterattack against the pathogenic bacteria [33]. The expressions of β-1,3-glucanase and chitinase were up-regulated in apples treated with *H. sinensis*. β-1,3-glucanases can active on fungal pathogens by degrading β-1,3/1,6-glucans in the fungal cell wall directly, making fungal cells susceptible to cell lysis [34]. After apples were treated with *H. sinensis*, the expression of GDSL esterase/lipase (GLP1) was up-regulated. When pathogenic bacteria were treated with recombinant GDSL esterase/lipase (GLP1), GLP1 directly disrupted the structures of fungal spore cell walls or membranes and interfered with the fungal infection process. 

### 4.5. Improve Plant Signal Transduction Ability

Plants are challenged by various abiotic and biotic stresses during growth and reproduction and even during post-harvest to sale. In such an environment, plants form complex signaling networks, and signal transduction proteins are the main regulators of plant signaling pathways [35]. After apples were treated with H. sinensis, the expression of signal transduction-related proteins was up-regulated, which resulted in the expression of defense-related genes and improved the stress resistance of apples. After H. sinensis treatment, G-type lectin S receptor-like serine/threonine-protein kinase, cell wall-associated receptor kinase-like 20 (WAKL20), LRR receptor-like serine/threonine-protein kinase At2g16250, LRR receptor-like serine/threonine-protein kinase ERL2, and leucine-rich repeat protein 2 showed up-regulated expression because their different domains responded to different external signals. When cell surface receptor-like protein kinases sense extracellular signals, they transmit the signals to intracellular signal response molecules, causing a series of intracellular responses, such as changes in intracellular Ca^2+^ levels, protein phosphorylation cascades activation, etc., thereby activating genes and proteins involved in stress regulation and cell protection [36].

### 4.6. Antimicrobial-Related Activity

After apples were treated with H. sinensis, the expression of proteins with antimicrobial activity was up-regulated, such as the thaumatin-like protein (TLP), basic 7S globulin (BG7S), ribosome-inactivating protein (LTP), non-specific lipid transfer protein, and Snakin-1 (SNI). TLP belongs to the pathogenesis-related protein-5, is rich in cysteine residues that promote protein stability, and also contains five additional conserved amino acids—arginine (R), glutamic acid (E), and three aspartate residues (D)—known as the REDD motif, and this structure has been implicated in the antifungal activity of TLP [37]. BG7S has stress response, antimicrobial activity, and hormone receptor-like activity. For example, BG7S in tomatoes can inhibit a cell wall-degrading enzyme from *Aspergillus aculeatus* to defend against pathogens [38]. LTP is an rRNA N-glycosylase evolved in plants, which can inactivate ribosomes of pathogenic bacteria and inhibit their protein synthesis, thereby responding to pathogen infection [39].

### 4.7. Substance-Related Transport

After apples were treated with H. sinensis, the expression of proteins involved in the transport of resistant substances and key signal molecules was up-regulated, such as the ABC transporter and the inorganic phosphate transporter. ABC transporters act on the cell membrane to transport compounds and participate in various biological processes. The ABC transporter G34 (ABCG34) in *Arabidopsis thaliana* mediates the secretion of the phytoalexin camalexin against *Alternaria brassicicola* infection [40]. Phosphorus is involved in various key metabolic pathways in plants, such as energy metabolism, signal transduction, and enzyme regulation [41].

### 4.8. Protein Processing

The plant immune system must rapidly identify and alter relevant signaling pathways and post-translational modifications to respond to different pathogens and infection processes. Post-translational modifications of host proteins are powerful regulators during plant immunity. The endoplasmic reticulum (ER) is an important subcellular organelle that acts as a checkpoint for protein folding and plays a crucial role in ensuring the correct folding and maturation of newly secreted and transmembrane proteins [42]. After apples were treated with H. sinensis, the expression of transporter SC16B in the protein-processing pathway in the endoplasmic reticulum was up-regulated. SC16B transported proteins on the ribosome to the endoplasmic reticulum for processing. After the protein entered the endoplasmic reticulum, the proteins responsible for protein processing and folding, such as protein luminal binding protein BiP5, DnaJ proteins P58IPK and ERDJ3B, and calnexin CNX1, were all up-regulated. Finally, the protein OS9 and transporter SC16B in the ERAD system responsible for the degradation of misfolded proteins were up-regulated, wherein the protein OS9 could recognize the unique asparagine-linked glycans on the folded proteins; then, the transporter SC61B converted the misfolded proteins from the ER retrogrades into the cytoplasmic matrix for degradation [43,44,45].

The overall molecular mechanisms of *H. sinensis*-induced enhancement of apple resistance could be summarized as follows (Figure 5): (1) *H. sinensis* treatment increased the expression of apple redox-related genes and proteins and catalyzed the release of toxic oxidation products under cellular stress, which are involved in hydrolysis and participate in the electron transfer in the redox process, to maintain the redox homeostasis of plant cells and protect the cells; (2) *H. sinensis* treatment increased the expression levels of resistance-related substances in apples and the key genes in the synthesis of resistance-related substances. The expressions of proteins, such as flavonoids, jasmonic acid, salicylic acid, and anthocyanins, stimulated the defense responses in apples; (3) *H. sinensis* treatment increased the expressions of key receptor proteins and key regulators in the PTI and ETI pathways in the plant immune system and improved the defense of apples against pathogens; (4) *H. sinensis* treatment increased the expression of proteins related to the degradation of pathogen cell walls in apples, such as β-1,3-glucanase and chitin; (5) *H. sinensis* treatment increased the expressions of genes and proteins associated with sensing external environmental stress signals and cell signal transduction during defense in apples and strengthened the relationship between plants and the environment. (6) *H. sinensis* treatment induced the expressions of some proteins with antifungal and antibacterial activities in apples, which directly attacked the pathogens; (7) *H. sinensis* treatment up-regulated the expressions of transporters of defense-related substances in apples and enhanced the response of apples to external stress; (8) *H. sinensis* treatment up-regulated the expression of protein-processing-related proteins in apples, improved the processing efficiency of proteins, and improved the resistance of apple.

## 5. Conclusions

In this study, we analyzed the major differential genes in apples induced by the antagonistic yeast *H. sinensis*. The results showed that the main defense mechanisms involved in differentially expressed proteins were scavenging active oxygen species, improving plant resistance and synthesis of resistant substances, improving plant disease resistance, cell wall degradation, cell signal transduction, antibacterial activity, transport of defense-related substances, and protein processing. These DEPs are related to fruit resistance, and their up-regulated expression indicates that *H. sinensis* can effectively induce and improve apple resistance. The molecular mechanism of *H. sinensis* controlling the postharvest disease of apples was revealed, which provides theoretical reference for the subsequent commercial application of the antagonistic yeast.

## Figures and Tables

**Figure 1 foods-12-02637-f001:**
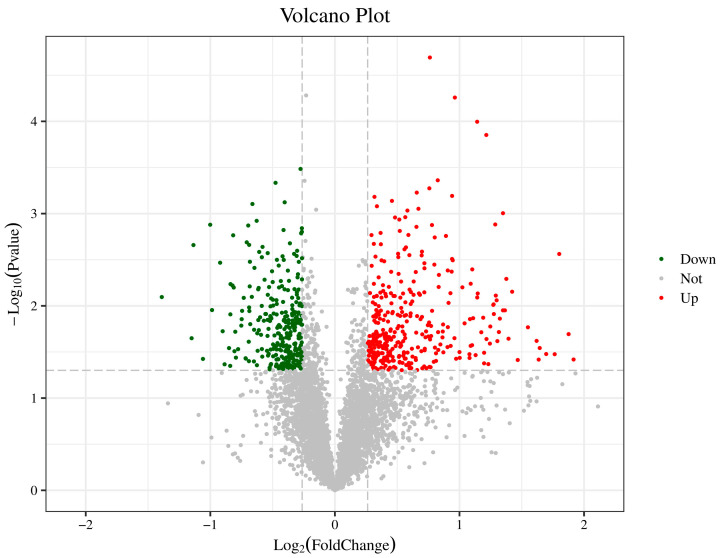
Volcano map of DEPs in apples of the control and yeast-treated groups. The green dots in the figure represent down-regulated DEPs, the red dots represent up-regulated DEPs, and the black dots represent non-DEPs.

**Figure 2 foods-12-02637-f002:**
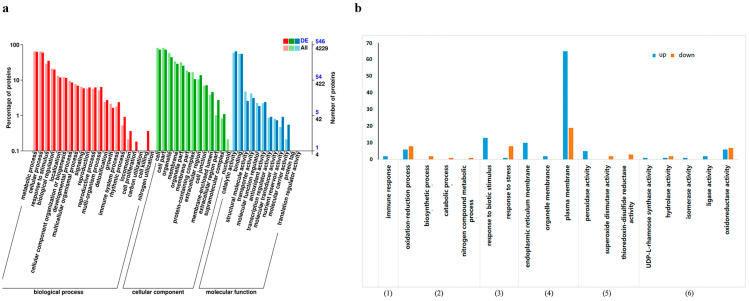
GO secondary annotation classification statistics of DEPs (**a**) and GO tertiary enrichment analysis of some DEPs (**b**) ((1)–(6) represent the six selected secondary pathways).

**Figure 3 foods-12-02637-f003:**
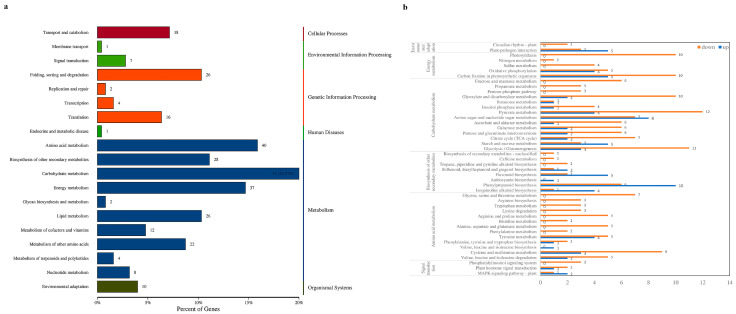
KEGG secondary annotation classification statistics map of DEPs (**a**) and KEGG tertiary enrichment analysis map of some DEPs (**b**).

**Figure 4 foods-12-02637-f004:**
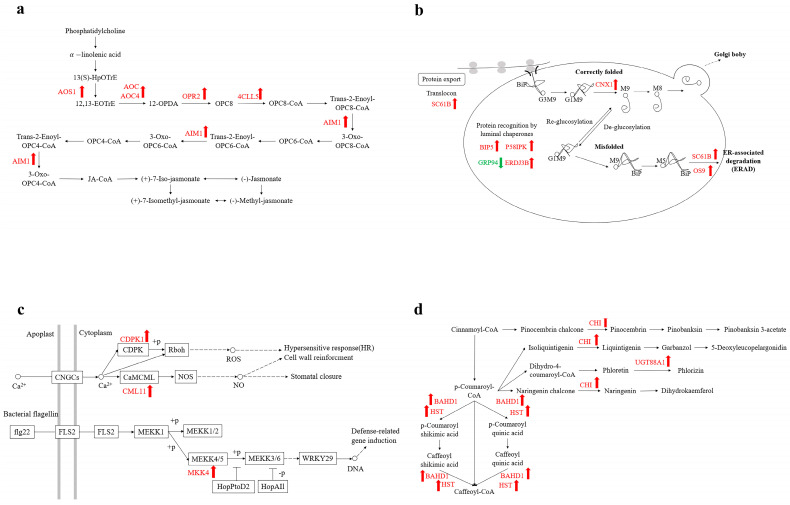
The KEGG pathway related to the *H. sinensis*-induced enhancement of apple resistance. (**a**) Jasmonic acid biosynthesis pathway; (**b**) protein processing in the endoplasmic reticulum; (**c**) plant-pathogen interaction; (**d**) flavonoid biosynthesis pathway.

**Figure 5 foods-12-02637-f005:**
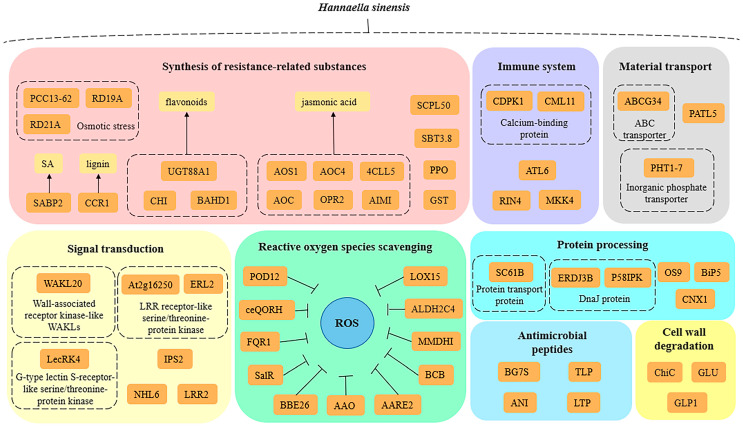
Schematic illustration of overall molecular mechanisms of *H. sinensis*-induced apple resistance based on proteomic results.

**Table 1 foods-12-02637-t001:** Functional classification of key DEPs related to the resistance mechanisms.

Gene ID	Protein Name	Fold Change	Description
**Oxidative stress**			
MD05G1168400	ceQORH	2.35	Chloroplast envelope quinone oxidoreductase homolog
MD08G1005500	AARE2	2.26	Acylamino-acid-releasing enzyme 2
MD15G1328900	AAO	2.21	L-ascorbate oxidase
MD00G1190800	POD12	1.88	Peroxidase 12
MD12G1218000	LOX15	1.78	Probable linoleate 9S-lipoxygenase 5
MD15G1187300	ALDH2C4	1.7	Aldehyde dehydrogenase family 2 member C4
MD02G1028600	BCB	1.68	Blue copper protein
MD05G1334200	FQR1	1.65	NAD(P)H dehydrogenase (quinone) FQR1
MD06G1077500	SalR	1.46	Salutaridine reductase
MD03G1240100	MMDHI	1.38	Malate dehydrogenase, mitochondrial
**Improving plant resistance**			
MD05G1320800	PPO	2.44	Polyphenol oxidase latent form, chloroplastic
MD06G1003300	CCR1	2.12	Cinnamoyl-CoA reductase 1
MD16G1149300	PCC13-62	2.00	Desiccation-related protein PCC13-62
MD02G1273000	BAHD1	1.85	BAHD acyltransferase At5g47980
MD05G1184300	GST	1.8	Probable glutathione S-transferase
MD16G1027500	SCPL50	1.68	Serine carboxypeptidase-like 50
MD16G1071800	HTH	1.57	Protein HOTHEAD
MD15G1078300	RD21A	1.53	Cysteine proteinase RD21A
MD01G1118100	CHI	1.7	Chalcone-flavonone isomerase
MD15G1194300	SKIP	1.52	SNW/SKI-interacting protein
MD10G1066900	RD19A	1.48	Cysteine protease RD19A
MD00G1035000	SBT3.8	1.48	Subtilisin-like protease SBT3.8
MD02G1162200	SABP2	1.47	Salicylic acid-binding protein 2
MD07G1306900	FGT	1.45	Anthocyanidin 3-O-glucosyltransferase 2
MD04G1139800	PHOS32	1.43	Universal stress protein PHOS32
MD06G1085500		1.37	Ribosome-inactivating protein bryodin II
MD17G1224900	HST	1.64	Shikimate O-hydroxycinnamoyltransferase
MD07G1280800	UGT88A1	1.33	UDP-glycosyltransferase 88A1
MD00G1038400	AOS1	1.25	Allene oxide synthase 1, chloroplastic
MD16G1047500	AOC	1.48	Allene oxide cyclase, chloroplastic
MD15G1402700	OPR2	1.44	12-oxophytodienoate reductase 2
MD15G1003500	4CLL5	1.21	4-coumarate—CoA ligase-like 5
MD15G1181900	AIM1	1.48	Peroxisomal fatty acid beta-oxidation multifunctional protein AIM1
**Improving plant disease resistance**			
MD05G1172400	RIN4	1.53	RPM1-interacting protein 4
MD16G1202100	CML11	2.46	Calmodulin-like protein 11
MD02G1257300	CDPK1	1.6	Calcium-dependent protein kinase 1
MD09G1157600	MKK4	1.26	Mitogen-activated protein kinase kinase 4
MD15G1377900	ATL6	1.51	E3 ubiquitin-protein ligase ATL6
**Pathogen cell wall degradation**			
MD15G1037300	ChiC	2.63	Class V chitinase
MD11G1107600	GLU	2.15	Glucan endo-1,3-beta-glucosidase
MD17G1151800	GLIP1	2.93	GDSL esterase/lipase 1
**Signal transduction**			
MD03G1020000	LecRK4	2.68	G-type lectin S-receptor-like serine/threonine-protein kinase LECRK4
MD01G1139800	WAKL20	1.62	Wall-associated receptor kinase-like 20
MD09G1187500	NHL6	1.47	NDR1/HIN1-like protein 6
MD08G1031300	At2g16250	1.46	Probable LRR receptor-like serine/threonine-protein kinase At2g16250
MD10G1102700	LRR2	1.43	Leucine-rich repeat protein 2
MD16G1004400	ERL2	1.42	LRR receptor-like serine/threonine-protein kinase ERL2
MD00G1040900	IPS2	1.4	Inositol-3-phosphate synthase
**Antibacterial activity**			
MD04G1064400	TLP	2.11	Thaumatin-like protein
MD11G1287900	BG7S	1.94	Basic 7S globulin
MD04G1173600	LTP	1.59	Non-specific lipid-transfer protein
MD08G1089100	SN1	1.37	Snakin-1
**Substance transport**			
MD14G1018100	ABCG34	2.16	ABC transporter G family member 34
MD13G1213400	PHT1-7	1.87	Probable inorganic phosphate transporter 1–7
MD15G1406000	PATL5	1.7	Patellin-5
**Protein processing**			
MD03G1297000	ERDJ3B	1.39	DnaJ protein ERDJ3B
MD03G1115300	SC61B	1.28	Protein transport protein Sec61 subunit beta
MD12G1215200	BIP5	1.21	Luminal-binding protein 5
MD04G1100100	P58IPK	1.27	DnaJ protein P58IPK homolog
MD14G1144700	CNX1	1.28	Calnexin homolog 1
MD02G1018200	OS9	1.32	Protein OS-9 homolog

## Data Availability

Data are contained within the article.

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
