# Peer review of "TMT-Based Proteomic Analysis of Hannaella sinensis-Induced Apple Resistance-Related Proteins"

_foods, 2023, doi:10.3390/foods12142637_

Round 1

Reviewer 1 Report

Dear Authors,

In this study, authors performed proteomics analysis based on TMT technology in apple treated with H. sinensis and treated with sterile saline to explore the key differential proteins in apple treated with H. sinensis, and to preliminarily elucidate the molecular mechanism of H. sinensis-induced enhancement of apple resistance. Considering the research on the molecular mechanism of antagonistic yeasts to control apple postharvest diseases is not comprehensive enough, this manuscript is important.  However, the manuscript has some deficiencies that do not make it publishable in its current state. It is written in a chaotic way. The manuscript needs substantial changes and additions to aspire to be published. The form of the English language also needs to be improved.

Abstract

The abstract 's structure is somewhat unbalanced: the part on how the study was conducted is very lengthy, while reporting little information on the results obtained. This part needs to be completely rewritten. The purpose of the abstract is to briefly illustrate the objective and the salient concepts of the research.

Introduction

The introduction is some verbose. Furthermore, it reports many concepts of a general nature related to predict H. sinensis induction increased the activities of apple resistance-related enzymes PPO, POD, APX, SOD and PAL that has very little to do with what is the objective of this study.  The introduction should show the current situation for estimating TMT-based proteomic analysis of Hannaella sinensis-induced apple resistance-related proteins, aslo be limited to highlighting the limitations and advantages that Hannaella sinensis-induced apple resistance-related proteins can achieve with the new method. The objective of the study (and consequently the title of the paper) should be better defined. In the near future (especially in the light of the climate change and the physiology of the Hannaella sinensis-induced apple resistance-related proteins) the resistance characteristics required of the apple cultivars could also be different. These concepts need to be explained better.

Materials and methods

The materials and methods should be written more clearly. Some methodological aspects need to be better explained. In particular, information must be provided regarding the apple, vegetative productive conditions and health status of the trees where apples are harvested, as well as the agronomical management of the experimental field. Also, information about which apple variety was used in the experiment should be given. It is necessary to report the dates in which the various steps of the experiment were carried out (this part is unclear). Furthermore,  the environmental conditions of the orchard must be indicated (e.g. temperature, relative humidity, hours of light, etc.) and the daily water and nutrient supply to the plants.

Results 

Many parts of the results are written in a very articulate, but somewhat muddled way. This way of presenting of results penalizes the quality of the paper. To enhance the results obtained and improve their comprehensibility to the reader, the authors should use a streamlined presentation of the results, trying to minimise the use of numerical values. 

Discussion

This session is quite long-winded. I In this session only the results obtained in this study need to be discussed (with the support of the existing literature on the subject). All superfluous parts (not directly connected with the results obtained) must be eliminated.

Conclusions

This section is incomplete in the study and has not been written. 

The form of the English language also needs to be improved.

Reviewer 2 Report

Qiya reported manuscript with title TMT-based proteomic analysis of Hannaella sinensis-induced apple resistance-related proteins. The articles has well written but need minor revision 

1. Why authors used different database names for annotations?

2. Figure 3. b is blur plz increase visuality

3. what KEGG secondary annotation tell us ?

4. why authors not validated any of protein showing down regulation or upregulation?

5. Why no conculision

NA

Round 2

Reviewer 1 Report

Accept in present form.

Accept in present form.